# Femtosecond Laser Cutting of Human Crystalline Lens Capsule and Decellularization for Corneal Endothelial Bioengineering

**DOI:** 10.3390/bioengineering11030255

**Published:** 2024-03-05

**Authors:** Olfa Ben Moussa, Louise Parveau, Inès Aouimeur, Grégory Egaud, Corantin Maurin, Sofiane Fraine, Sébastien Urbaniak, Chantal Perrache, Zhiguo He, Sedao Xxx, Oliver Dorado Cortez, Sylvain Poinard, Cyril Mauclair, Philippe Gain, Gilles Thuret

**Affiliations:** 1Laboratory Biology, Engineering and Imaging for Ophthalmology, BiiO, Faculty of Medicine, Health Innovation Campus, Jean Monnet University, 10 Rue de la Marandière, 42270 Saint-Priest-en-Jarez, France; olfabm@outlook.com (O.B.M.); zhiguo.he@univ-st-etienne.fr (Z.H.);; 2GIE Manutech-USD, 18 Rue Professeur Benoît Lauras, 42000 Saint-Etienne, France; 3Laboratoire Hubert Curien, Jean Monnet University, 18 Rue Professeur Benoît Lauras, 42000 Saint-Etienne, France; 4Ophthalmology Department, University Hospital, Avenue Albert Raimond, CEDEX 02, 42055 Saint-Etienne, France

**Keywords:** corneal bioengineering, femtosecond laser, decellularization, human anterior lens capsule, viable tissue-engineered endothelial keratoplasty

## Abstract

The bioengineering of corneal endothelial grafts consists of seeding in vitro cultured corneal endothelial cells onto a thin, transparent, biocompatible, and sufficiently robust carrier which can withstand surgical manipulations. This is one of the most realistic alternatives to donor corneas, which are in chronic global shortage. The anterior capsule of the crystalline lens has already been identified as one of the best possible carriers, but its challenging manual preparation has limited its use. In this study, we describe a femtosecond laser cutting process of the anterior capsule of whole lenses in order to obtain capsule discs of 8 mm diameter, similar to conventional endothelial grafts. Circular marks made on the periphery of the disc indicate its orientation. Immersion in water for 3 days is sufficient to completely remove the lens epithelial cells and to enable the seeding of corneal endothelial cells, which remain viable after 27 days of culture. Therefore, this method provides a transparent, decellularized disc ready to form viable tissue engineered endothelial grafts.

## 1. Introduction

The cornea is the outermost multi-layered transparent tissue of the eye. Its innermost layer—the corneal endothelium—is a monolayer of flat hexagonal cells adherent to their basal membrane called the Descemet membrane. These terminally differentiated cells precisely control corneal hydration and consequently corneal transparency [1]. In their physiological environment, these cells are deprived of significant proliferative capacity and therefore cannot regenerate if their population is highly damaged. The loss of endothelial function results in permanent corneal edema and profound decrease in visual acuity. If the patient is treated before the irreversible complications of corneal oedema occur (stromal fibrosis responsible for permanent opacities, often after many months of evolution), the restoration of endothelial function can allow the recovery of perfect transparency. Endothelial transplantation has revolutionized the prognosis of corneal pathologies for the past 15 years [2]. This procedure consists of replacing the pathological endothelium with the healthy one from a deceased donor.

Corneal endothelial diseases are frequent and dominated by three etiologies. The first etiology is Fuchs endothelial corneal dystrophy, a primary endothelial disease affecting 4 to 10% of adults in Western countries in varying degrees [3]. The second etiology is pertaining to the complications of crystalline lens surgery (the rate of endothelial decompensation is low, but cataract surgery is the most frequent surgery worldwide, with approximately 25 million eyes operated yearly worldwide) [4]. The third etiology is the endothelial dysfunction of a previous corneal transplantation. Today, approximately 50% of all corneal transplantations worldwide are associated with these endothelial diseases [5,6], and this proportion keeps increasing. Nevertheless, the number of corneal donors available is very limited. We have estimated the imbalance between supply and demand to be of 1 cornea available for 70 waiting patients worldwide, with huge disparities between countries [7].

The development of alternatives to corneal donation is imperative if we want to reduce the burden of corneal blindness. Cell and/or tissue therapy is considered as the most realistic proposition [8]. Kinoshita et al. demonstrated the feasibility of injecting a suspension of cultured corneal endothelial cells (CECs) and an adherence-promoting drug (a Rho-kinase inhibitor) into the anterior chamber of patients suffering from Fuchs dystrophy and bullous keratopathy, whose results are strictly comparable to endothelial grafts performed with donor corneas [9]. By employing this strategy, it is possible to culture enough cells from a cornea of a donor under 30 years old (as their cells possess significantly greater proliferative capacities in vitro than cells from older donors) to treat several patients.

An alternative approach to the injection of suspended cells is the bioengineering of endothelial grafts, also known as tissue-engineered endothelial keratoplasty (TEEK), which consists of seeding in vitro cultured endothelial cells on an ultra-thin, transparent, and cornea-compatible carrier [10].

TEEK effectively reproduces the endothelial graft performed with a donor cornea [11,12]. Its manufacturing process allows for precise control over the number of cells delivered to the patient, and the cells are in their physiological state (adherent to the support and forming a functional, contiguous monolayer). Surgeons are also proficient in this transplantation technique. The carrier must be permeable to water, gases, nutrients, and metabolic wastes and have biomechanical characteristics [13,14], allowing manipulation without damaging it during in vivo transplantation into the recipient [4].

A wide variety of carriers have already been considered, i.e., amniotic membrane [15], natural polymers (collagen I and IV, fibronectin, gelatin, laminin, the combination of laminin and chondroitin sulfate or fibronectin, collagen, and albumin) [16], silk fibroin [17], the posterior layer of human corneas not suitable for conventional grafts [18], or the anterior part of the human crystalline lens capsule [19]. They have been tested, mostly in vitro and less frequently in in vivo animal models [10]. Among these candidates, the anterior part of the crystalline lens capsule has demonstrated numerous advantages. It is a basement membrane produced by lens epithelial cells. Mainly composed of collagen IV and laminin, it is transparent, slightly curved (compatible with corneal posterior curvature), and has a mean thickness of 8 to 27 µm, increasing with age [20]. It can be decellularized with chemicals and used as a scaffold to cultivate corneal endothelial cells [21]. This tissue has excellent properties for corneal bioengineering because it sustains endothelial cell phenotype [20]. Previously, we reported the preclinical study in rabbits of a 6 mm diameter TEEK composed of an anterior lens capsule disc (LCD) seeded with human corneal endothelial cells [22]. The LCD was a surgical waste product from a femtosecond laser-assisted cataract surgery. In this surgical procedure, the maximum diameter of the capsular cut was 6 mm, whereas the typical diameter of an endothelial graft is 8 mm, corresponding to a surface area 1.78 times larger and providing a greater pool of endothelial cells, likely to ensure a prolonged survival of the graft.

In this study, we reported the use of femtosecond laser (FsL) cutting process to obtain an 8 mm diameter LCD from an intact whole human lens, along with a chemical-free decellularization method that allowed the seeding of human corneal endothelial cell (hCECs). This process aimed to produce a perfectly safe tissue-engineered endothelial keratoplasty (TEEK).

## 2. Materials and Methods

This study took place at the Biology, Engineering and Imaging for Ophthalmology (BiiO) Laboratory located at the Faculty of Medicine Jacques Lisfranc, 10 Rue de la Marandière, 42000 Saint-Etienne.

### 2.1. Human crystalline lens

Fifteen lenses were retrieved during corneal donation (University Hospital of Saint-Etienne, France). Until now, the lens has been a surgical waste product extracted from the eye at the time of corneal procurement and discarded. We set up a protocol to use it as a new tissue. In the absence of opposition to donation and use of the tissue for scientific purposes, it was collected with the hospital coordination for organ and tissue procurement. An authorization for procurement for scientific purposes had been obtained from the Biomedicine Agency to procure the crystalline lenses, in addition to the corneas. Particular care was taken to preserve the lens capsule (no direct pinching). The age of donors was 69 ± 10.7 (mean ± standard deviation). Lenses were stored in 20 mL of organ culture medium (CorneaMax; Eurobio, Les Ulis, France) at 31 °C for up to 163 days (96 ± 50) before processing, in order to assess the possibility to use very long-term stored lenses. All samples used in this study were handled in accordance with the tenets set forth in the Declaration of Helsinki.

### 2.2. Femtosecond Laser Cutting

Lenses were enclosed in a custom-made holder designed to facilitate the centration of the laser cutting (Figure 1A). They were then cut using an experimental femtosecond laser (FsL) platform. The laser source was a Satsuma HP (Amplitude, Bordeaux, France) operating at a central wavelength of 1030 nm with a pulse duration of 328 fs. The laser pulses were sent through a Galvanometer scanner (Scanlab, Puchheim, Germany) and focused by a f-theta focusing lens. The laser focus spot was a Gaussian-shaped spot with a diameter of 8 µm at 1/e^2^ and an inter-spot distance of 5 µm. The laser pulse energy was 4 µJ. The cutting of the lens capsule was performed to obtain an LCD with 3 asymmetrical circular marks (2 in a row and 1 faraway). This pattern, described initially for endothelial graft preparation, was necessary for guarantying the right orientation of the tissue (only one side is intended to be seeded with cells of interest) [23]. To achieve this, the laser was scanned following a circle-like pattern in the *x*-*y* plane with a repetition rate of 250 kHz, yielding a 5 µm distance between the irradiation sites along the circle periphery as depicted in Figure 1. This circular pattern was repeated vertically (along the *z*-axis) with steps of 2 µm on a distance of 4.6 mm centered on the capsule surface (Figure 1B,C). This tubular irradiation pattern with a high density of irradiation spots permitted to make sure that the thin capsule membrane was correctly cut.

### 2.3. Anterior Lens Capsule Dissection

The LCD were dissected using micro-chirurgical materials under a binocular microscope (SZ61, Olympus, Tokyo, Japan). It was stained with 0.06% trypan blue (VisionBlue, DORC, Zuidland, The Netherland) to enhance the visualization and then carefully removed using toothless forceps by detaching the border of the disc from the rest of the lens. The discs were immediately spread on a microscopic slide with a drop of sterile water in order to assess their morphology.

### 2.4. Decellularization

The objective of decellularization is to thoroughly remove cellular residues while maintaining LCD integrity, in order to allow recellularization with CECs. LCD were decellularized by immersion in 4 mL of water supplemented with 1/100 e of antibiotic antimycotic (15240-062, Gibco, Saint-Louise, MO, USA), under constant agitation in a sterile borosilicate glass bottle (11553542, Fisher Scientific, Loughborough, UK) at room temperature. The goal of this step is to completely remove every cell or cell debris of each capsule. LCD was agitated for 1 or 3 days in order to determine the optimal duration allowing the complete removal of lens epithelial cells. Three LCD were decellularized in both groups, and three additional LCD were used as controls and analyzed immediately after dissection (time between death and cut of 114 ± 36 days). At the end of incubation, the presence of cytoplasmic and nuclear debris were assessed using double-staining with DiOC6 and DAPI as we previously reported [24]. DiOC6 (DiOC6(3) (3,3′-dihexyloxacarbocyanine iodide, D-273; Molecular Probes, Thermo Fisher Scientific, Waltham, MA, USA) is a cytoplasmic stain of the organelles and DAPI (D1306, Life Technology, Carlsbad, CA, USA) is a DNA stain. The LCD were incubated for 15 min at room temperature in a solution containing 2.5 µg/mL of DiOC6 and 2 µg/mL of DAPI in DPBS.

### 2.5. PCR Analysis

PCR was performed on lens capsules at D0, D3, and D6 of the decellularization process. Before each extraction step, the samples were rinsed twice with sterile water to remove any potential nuclear debris.

*Extraction and purification of genomic DNA.* A total of four different samples were used for DNA quantification: a positive control with a freshly peeled capsule without any shaking steps, and three other capsules shaken to be decellularized for 3 days or 6 days. Next, gDNA was extracted from the capsules, following all the steps recommended by the supplier of the Minikit PureLink genomic DNA kit and published by Andrea Catalina Villamil Ballesteros [25]. Briefly, to prepare the lysates, capsules were placed in 180 µL PureLink genomic digestion buffer and 20 µL proteinase K to remove lipids and digest proteins. Samples were then incubated in a water bath at 55 °C for 1 h, with stirring every 10 min. Samples were then centrifuged at 13,000 rpm for 3 min at RT. A 20 µL volume of RNAse was added to the supernatant for 2 min at RT followed by the addition of 200 µL of purelink Genomic Lysis/Binding Buffer. After vortexing, 200 µL of pure ethanol was added to facilitate DNA binding to the column. The DNA was rinsed with 500 µL of Wash Buffer 1 and 2 at 10,000× *g* for 1 min and at maximum speed for 3 min, respectively. DNA was eluted for 1 min by adding 1.5 mL sterile water containing 25 µL of PureLink Genomic Elution Buffer, followed by centrifugation at maximum speed at RT. All column and/or collection tube changes were carried out according to the manufacturer’s instructions. All extractions were stored at −80 °C until use.

*DNA quantification.* After the DNA extraction, the DNA samples were prepared following the protocol below. Amplification was performed on all samples on the same day. Total DNA was detected as described above. The amplification step was performed using the “Luna, Universal Probe qPCR Master Mix” kit (#M3003X, New England, BioLabs, Ipswich, UK), which is compatible with DNA samples prepared through typical nucleic acid purification methods. Volumes and dilutions were added according to the supplier’s instructions [26].

The thermocycling protocol was followed as indicated, i.e., initial denaturation at 95 °C for 60 s, followed by 50 cycles of denaturation at 95 °C for 15 s, and annealing extension at 58 °C for 30 s. DNA was detected by targeting a GAPDH housekeeping gene (personalized by Eurogentec). The sequences were as follows: forward primer: 5′-GAA-GGT-GAA-GGT-CGG-AGT-3′; reverse primer: 5′-GAA-GAT-GGT-GAT-GGG-ATT-TC-3′.

Positive controls composed of an immortalized human corneal epithelial cell culture (CRL-11135) were also tested on the same plate at 3 different dilutions: 1/20, 1/200 and 1/2000.

### 2.6. Corneal Endothelial Cells Culture and TEEK Reconstruction

Human corneas were obtained from an 86-year-old donor with a short time between death and procurement (9 h). Corneal endothelial cells were cultured according to previously published protocol [27]. Briefly, the endothelium was stained with 0.4% trypan blue (Eurobio Scientific, Paris, France) and mechanically peeled off from the cornea. The Descemet membrane was submitted to enzymatic digestion by incubation in OptiMEM-I (11058-021; Life Technologies Corp., Carlsbad, CA, USA) supplemented with 1 mg/mL collagenase A (10103586001; Sigma-Aldrich, Mannheim, Germany) at 37 °C for 16 h. After digestion, cells were washed with OptiMEM-I and then seeded in one well of a 24-well plate (190 mm^2^/well) coated with i-Matrix-511 (892012; Nippi, Incorporated, Tokyo, Japan). Cells were cultured with OptiMEM-I supplemented with 8% FBS, 10 µM SB203580 (S1076, SelleckChem, Houston, TX, USA), 1 µM SB431542 (S1067, Selleckchem), 5 ng/mL epidermal growth factor (EGF, A42554, Thermo Fisher), 20 µg/mL ascorbic acid (A5960; Sigma-Aldrich), 200 mg/l calcium chloride, 0.08% chondroitin sulphate (034-14612, Wako PureChemical Industries, Ltd., Osaka, Japan), and 1/200 of antibiotic antimycotic 100X (15240-062, Gibco). A 10 µM quantity of Y-27632 Rho-kinase inhibitor (S1049, SelleckChem) was added in the medium for only the first 24 h after cell seeding. Cells were cultured in an incubator with a humidified atmosphere at 37 °C in 5% CO_2_, and the medium was replaced with fresh medium every week. When the cells reached confluency in about 28 days, they were rinsed with Ca^2+^ and Mg^2+^-free DPBS, detached with TrypLE (A1217701, Thermo Fisher) for 15 min at 37 °C, and passaged at a 1:2 ratio. For this experiment, cells were used after eight passages. They retained a typical endothelial phenotype (hexagonal and regular), and the endothelial cell density was 1414 cells/mm^2^. For TEEKs’ assembly, the LCD were spread in 24-well plates coated with FNC coating mix (Athena ES, 0407) in order to increase their adherence to the plastic. The LCD was then coated with 2.5 µg/mL i-Matrix-511(892012; Nippi Incorporated). As preconized by the supplier, the i-Matrix-511 was left in wells for 1 h at 37 °C and entirely aspirated and replaced by a seeding medium, without drying steps. Cells were then harvested and seeded on pre-coated LCD at 2000 cells/mm^2^. The culture plate was kept in an incubator with a humidified 5% CO_2_ atmosphere at 37 °C, and the medium was replaced with fresh culture medium once per week for 4 weeks. The 4-week period was chosen because it corresponds to the maximum storage time for corneal grafts in organoculture (4 to 5 weeks) and would allow easy organization of the allocation of TEEKs to patients and their transport to the operating room.

### 2.7. Cell Viability Assessment

To measure the viability of TEEKs and controls, a triple staining with Hoechst–Ethidium homodimer–Calcein-AM (HEC) was used [28]. A solution of 5 µg/mL Hoechst (B2261, Sigma, Saint Quentin Fallavier, France), 4 µM Ethidium (E) (FP-AT758A, Interchim, Montlucon, France), and 4 µM Calcein-AM (FP-FI9820, Interchim, Montlucon, France) was prepared in OptiMEM-I, and 300 µL of the solution was used per well. The solution was incubated for 45 min at room temperature (RT).

### 2.8. Immunolabeling of Endothelial Markers

We used a protocol optimized for the immunolabeling of flat-mounted endothelium [29]. Briefly, TEEKs were rinsed in Dulbecco’s Phosphate Buffer Saline (DPBS) containing Ca^2+^ and Mg^2+^ (Gibco, 14040-133, Life Technologies Corporation, USA) and then fixed in pure methanol at RT for 30 min. The TEEKs were then rehydrated in DPBS and incubated in blocking buffer (DPBS, 2% bovine serum albumin, 2% goat serum) for 30 min at 37 °C. The primary antibodies, diluted at 1/500 in blocking buffer, were incubated with TEEKs at 37 °C for one hour. After three rinses with DPBS, the secondary antibodies, diluted at 1/500, and DAPI diluted at 2 µg/mL in blocking buffer were incubated with cells at 37 °C for one hour. After three rinses with DPBS, the TEEKs were immersed in Fluoromount-GTM mounting medium (00-4958-02, Invitrogen, Carlsbad, CA, USA) to protect the fluorochromes (Alexa Fluor™ 488, Alexa Fluor™ 555 and DAPI). Image acquisition is performed with an epifluorescence inverted microscope (IX81, Olympus, Tokyo, Japan) with the CellSens software (Soft Imaging System GmbH, Olympus). Two primary antibodies, anti-human CD166 (559260; BD Pharmingen, BD biosciences) and anti-human CD56 (MAB24081, R&D systems), were used as marker of human CECs [1,22,29]. CD166 is a basolateral transmembrane protein involved in cell–cell adhesion. NCAM is a specific endothelial marker localized on the lateral membrane of CECs [30,31].

### 2.9. Imaging Techniques

An optical coherence tomography (SS-OCT ATR200, OCTG-NR Scanner, OCT-LK4, Thorlabs, Newton, NJ, USA) was used to observe lens before and after FsL cutting, and a macroscope imaging (macro-zoom microscope) (MVX10; Olympus, Tokyo, Japan) at ×1.25 magnification after cutting. These images were used to check that the laser cut was centered and complete over 360°.

For decellularization characterization, images of the whole LCD were obtained with a macroscope equipped with fluorescence at ×1.25 magnification. High-magnification images were also collected at ×12.6 magnification from 5 different zones (4 peripheral and 1 central zone). To detect debris with a higher resolution, a confocal microscope (Fluoview FV1200; Olympus) was used at ×20 and ×60 magnification.

Transparency was assessed using a custom-made noninvasive device called transparometer [32]. LCD were first rinsed in water for 4–7 days to attenuate the blue staining and photographed on a backlit chart.

Immunolabeling of TEEKs was imaged with an epifluorescence inverted microscope (IX81, Olympus) in multiple fields, at ×10 and ×40 magnification.

### 2.10. Endothelial Cell Density Measurement

Endothelial cell density of TEEKs was measured by segmenting stained cell nuclei (Hoechst or DAPI) with the Stardist Plugin [33,34] of ImageJ (version 1.54), on images taken with an inverted microscope at ×10 magnification or macro-zoom macroscope at ×6.3 magnification.

## 3. Results

### 3.1. Femtosecond Laser Cutting of Lens Capsule Discs

Figure 2 showed OCT images of lenses before and after FsL cutting. By using the custom-designed holder, the samples remained stationary during laser cutting, ensuring correct centering of the laser pattern. The laser parameters allowed a reproducible cutting of an 8 mm diameter with three circular marks for all LCDs (Figure 3A). Whether the lenses were swollen (lens 1) or in an almost physiological state (lens 2), laser cut remained unaffected (Figure 2). Figure 3B, all LCDs were transparent, with no evidence of line deformation.

### 3.2. Decellularization Efficiency

Control LCD, which was not decellularized, exhibited epithelial cell remnants stained with DiOC6 (cytoplasm) and DAPI (nuclei) across the entire surface. Following a one-day decellularization, some cellular debris was observed in confocal images, scattered on the LCD. After 3 days, macroscope images revealed complete decellularization, as confirmed by confocal microscopy: neither cells nor cellular debris were visible on the capsule (Figure 4).

The minimal number of PCR cycles observed for the DNA amplification was 20 for the most concentrated control, i.e., 1/20 (and 28 and 30 for the positive controls at 1/200 and 1/2000 cell dilution). For non-decellularized capsules, DNA amplification was detected after 26 cycles (Figure 5A). For capsules decellularized for 3 and 6 days, DNA detection was observed at 22 and 21 cycles, respectively (Figure 5B,C).

### 3.3. Endothelial Cell Viability, Morphology, and Density of TEEKs

The CECs rapidly formed a uniform monolayer on the capsule surface and remained stable for 4 weeks. CEC viability was close to 100%. We observed no difference in viability between CECs adhering to the capsule and those adhering directly to the plastic of the culture well (Figure 6A,B).

Immunostaining for CD166 and NCAM confirmed that TEEK’s CECs had a phenotype consistent with native CECs (Figure 5C) [1].

The ECD of the four TEEKs after 4 weeks was 1798 +/− 137 cells/mm^2^ (1740, 1980, 1813 and 1659, respectively), while the control well was 1496 cells/mm^2^.

## 4. Discussion

In the therapeutic arsenal for advanced corneal endothelial diseases, endothelial keratoplasty using a donor cornea is the reference treatment. Unfortunately, the number of donors remains significantly lower than the demand [7], especially since these corneas are also essential for treating numerous other corneal diseases. The cell therapy developed by the Kyoto team (Pr Kinoshita, Koizumi, and Okumura) should soon reduce this dependence on donors [9,35]. Alongside the cell injection in suspension developed by this Japanese team, TEEKs should find their place thanks to their specific properties linked to the fact that they effectively reproduce the original corneal endothelium.

The newly developed TEEKs graft technique is expected to offer a viable alternative to conventional endothelial grafts, which currently represent half of all grafts used globally. This innovative method has the potential to replace around 50% of existing endothelial graft techniques.

Among the many possible supports for the assembly of TEEKs, we chose the crystalline lens capsule for its multiple well-demonstrated biological and biophysical properties but also with the perspective of a rapid transfer to patients [19,21]. The biocompatibility of the lens capsule is evident, and we have already performed the proof of concept of a functioning TEEK in rabbits by successfully transplanting TEEKs built with LCD (6 mm in diameter) and human corneal endothelial cells [22]. The limitation for large-scale clinical application is due to the lens capsule’s availability and the possibility of cutting discs of sufficient diameter. The recovery of surgical waste during cataract surgery (which is regulated similarly to living tissue donation such as femoral head or placenta) is the first interesting option because cataract surgery is the most common surgery in the world (at least 25 million eyes operated on each year). However, the capsule can only be recovered during laser-assisted surgeries (or another method of capsule cutting) that preserve an intact central disc (which is necessarily torn during manual surgery). Moreover, its average diameter is 5 mm, and its maximum diameter is 6 mm, well below the usual 8 mm of endothelial grafts. Although it is theoretically possible to perform a small surface endothelial graft [36,37], significant doubts persist regarding the long-term survival of these grafts with few cells. The second option is the use of whole crystalline lenses. This is the only possibility to obtain 8 mm diameter LCDs. Whatever the age of cornea donors, the crystalline lens has a slightly larger diameter of over 9 mm [38]. The retrieval of the crystalline lens for therapeutic purposes is not yet authorized and will require specific approaches from competent authorities. It is currently a surgical waste during enucleation or in situ cornea harvesting. In our population of donors with an average age of 72 years, the crystalline lens was still present in 70% of the eyes (internal data from our eye bank over 5 years and 3500 donations, with no tendency to decrease over time). Availability is therefore not an obstacle. Manual cutting of an 8 mm disc is uncertain and carries a significant risk of failure, whereas laser cutting is safe. We have therefore used an experimental femtosecond laser platform to perform the cut because none of the commercially available ophthalmic femtosecond laser platforms allow customized cutting on an isolated crystalline lens. The cutting pattern can be modified, as we have shown by adding three circular marks to indicate the orientation of the tissue. The crystalline support that we have created allows for the stabilization and the centering of the cut. The entire process can be integrated into a clinical batch production process. The cutting is performed without contact, on a crystalline lens placed in a sterile transparent box, in a clean room. This process showed a reproducible cut of 8 mm of diameter on LCD, which has a size similar to DMEK grafts (mostly 8 and exceptionally 9 mm) [39]. Given the cost of developing a dedicated femtosecond laser machine, we believe that it will be possible to centralize the preparation of lens capsules in a small number of centers, each of which will perform a large number of cuts. The crystalline lenses will then be retrieved in multiple hospitals and sent to these few specialist tissue banks.

We have also shown that it is possible to use crystalline lenses preserved in sterile conditions for several months. These lenses exhibit significant edema due to cellular necrosis but can be easily cut with a laser. We have not observed any degradation of the capsule over time. This result suggests that it is possible to collect lenses from several centers, to store them without damage, and organize the preparation of a large number of LCD at the same time to produce a batch of significant size. Our decellularization process is also simplified to facilitate transfer to the clinic. This chemical-free technique consisting of immersion in water is effective to remove every cellular and nuclear debris. After PCR amplification, DNA from epithelial cells was detected in capsules decellularized for 3 days and 6 days. These results indicate that the decellularization method removes cells but does not completely remove DNA content. For future applications and with the aim of translating this protocol to the clinical stage, we must add technical steps to remove DNA.

Finally, we have demonstrated our capability to seed hCECs on the LCD. These hCECs maintained viability for a period of at least 4 weeks and kept their density and endothelial morphology throughout the culture period. Maintenance of ECD is easy because it requires classic cell culture technique. For further study, the initial seeding density of CECs should be higher to enhance the final ECD on the TEEK graft, likely to improve graft survival in patients.

With this process, we are able to produce new human tissue that is ready for use. While the first application envisioned is the bioengineering of TEEK, other applications of decellularized LCD are possible, such as the bioengineering of grafts of retinal pigment epithelium, photoreceptors, or ganglion cells. Additionally, LCD could also be used to treat certain macular holes that have failed conventional surgeries [40].

## 5. Conclusions

The process we present allows the reproducible cutting of 8 mm diameter anterior LCD with orientation marks. The extremely simple decellularization process results in a perfectly transparent membrane. The seeding of hCECs on this carrier allows obtaining a TEEK that can be stored for weeks. The process is easily transferable to GMP standards to create tissue-engineered endothelial keratoplasty that exactly mimics endothelial grafts dissected from donor corneas.

## Figures and Tables

**Figure 1 bioengineering-11-00255-f001:**
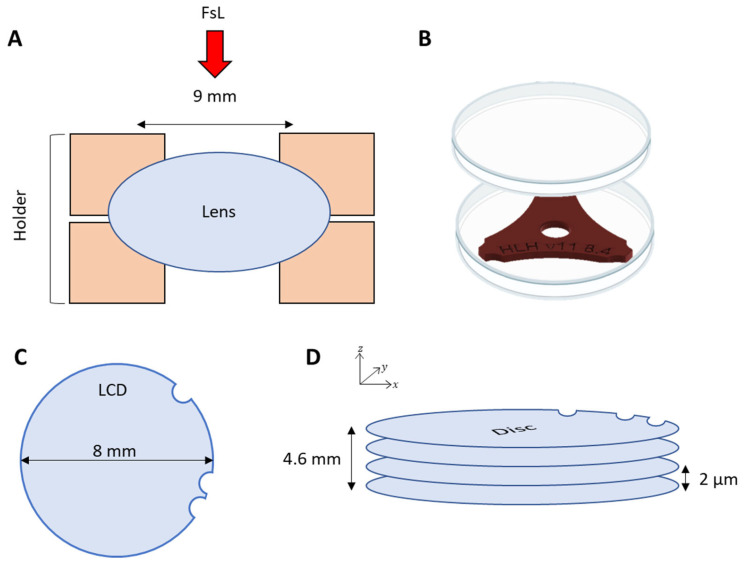
Lens holder principle and femtosecond laser pattern. (**A**) Schematics of the human lens holder used to maintain and cut the anterior lens capsule disc. The opening of the support, which is equal to 9 mm, allows a laser cutting of 8 mm in diameter. (**B**) Human lens holder included in a sterile Petri dish to maintain sterility. (**C**) FsL pattern with a disc of 8 mm and 3 asymmetrical circular marks (2 in a row and 1 faraway). (**D**) Graphic of the cutting parameters with an inter-disc distance of 2 µm and a total height cut of 4.6 mm.

**Figure 2 bioengineering-11-00255-f002:**
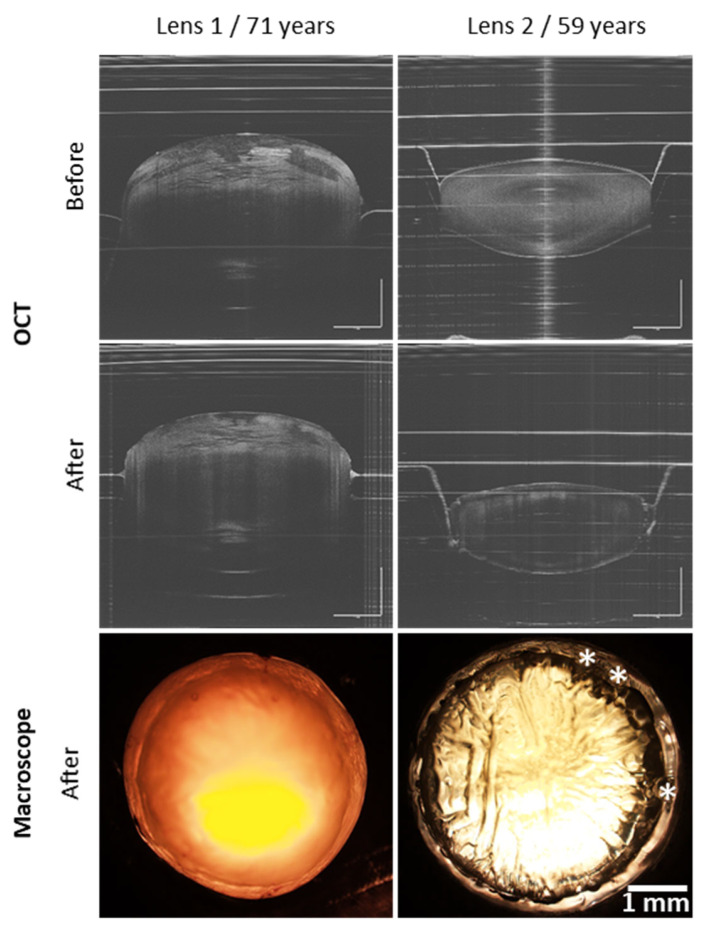
Representative examples of 2 lenses before and after femtosecond laser cutting of an 8 mm diameter disc. Optical coherence tomography (OCT) acquisition before and after femtosecond laser cutting showed good lens stability during the process. Retro-illuminated pictures allowed verification of the correct centration before manual dissection and showed the 3 circular marks (*).

**Figure 3 bioengineering-11-00255-f003:**
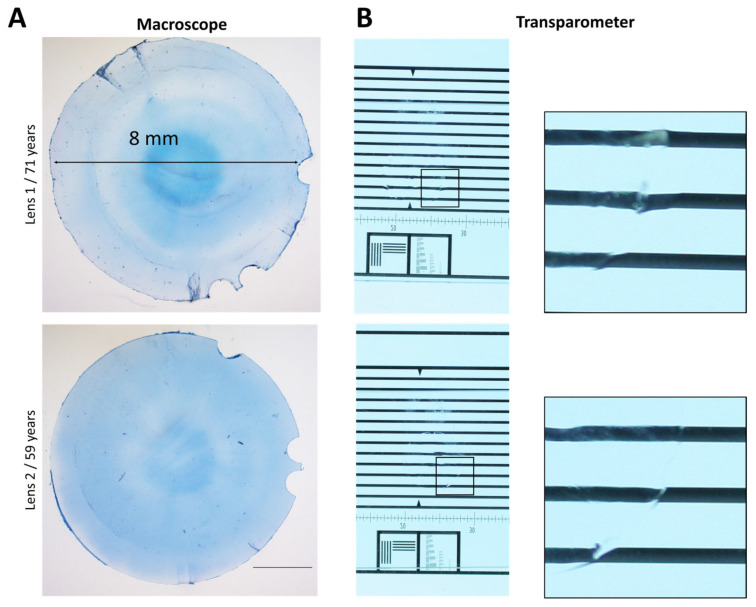
Human anterior lens capsule disc cut by femtosecond laser. (**A**) Low magnification images of the lens capsule disc stained by trypan blue to facilitate handling. (**B**) Transparency assessment using a backlit chart. A zoom showed the edge of each capsule. The discs were hardly visible.

**Figure 4 bioengineering-11-00255-f004:**
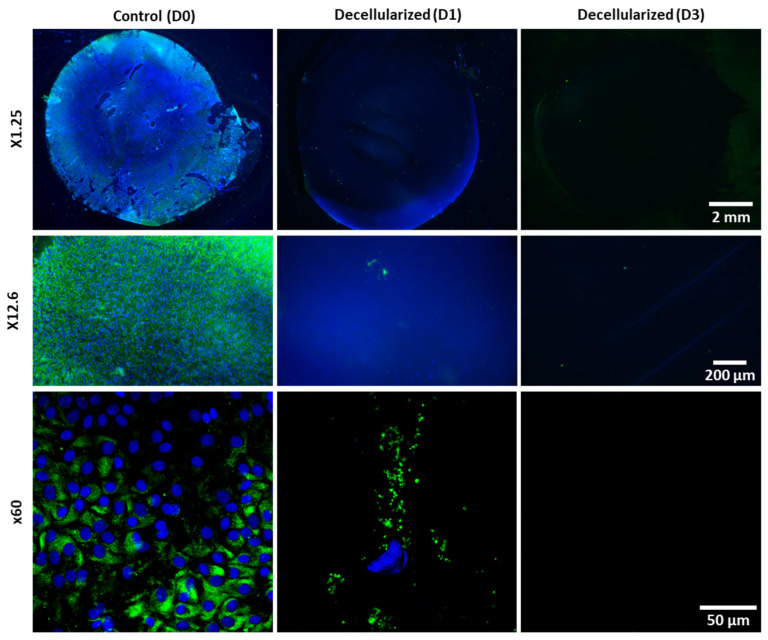
Efficiency of decellularization in water. Cells residues were stained with DiOC6 (cytoplasm, in green) and DAPI (nuclei, in blue) staining. Representative examples after 1 and 3 days of immersion in water, versus control. Images were taken at a magnification of ×1.25 for the whole capsule, ×12.6 for the zoom, and at ×60 for cell resolution.

**Figure 5 bioengineering-11-00255-f005:**
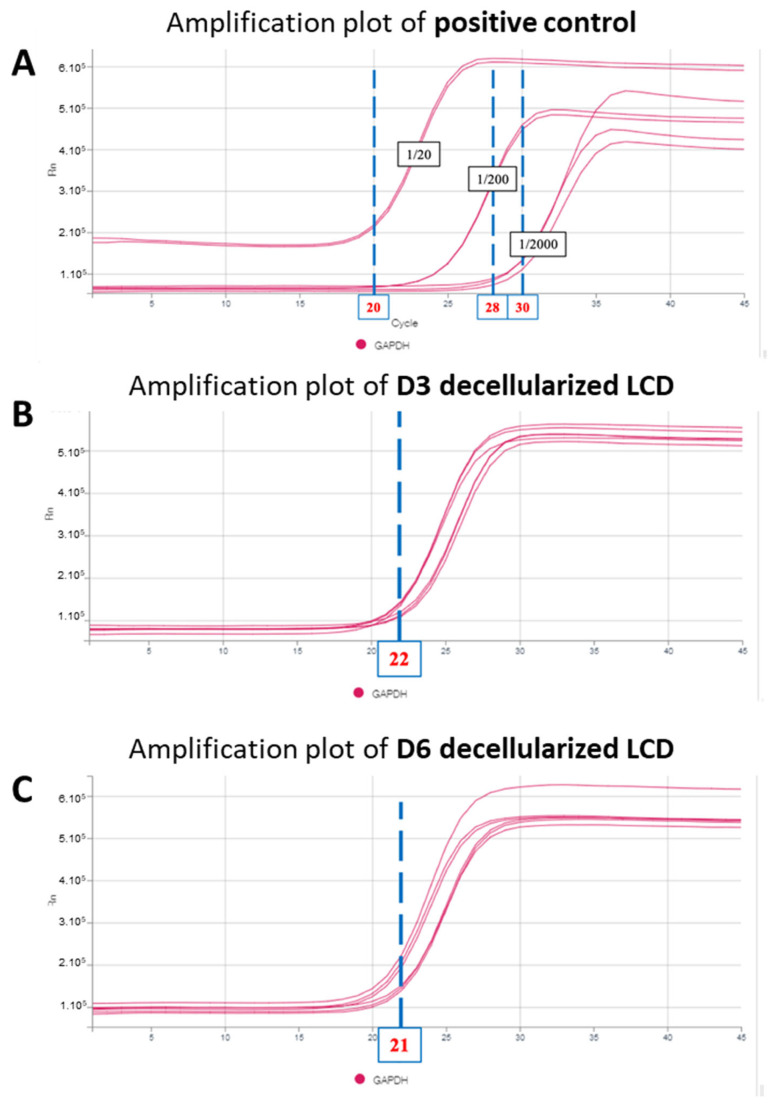
Quantification of DNA detected by PCR. (**A**) DNA amplification of positive samples at a concentration of 1/20, 1/200, and 1/2000. (**B**) DNA amplification of capsules decellularized for 3 days. (**C**) DNA amplification of capsule decellularized for 6 days.

**Figure 6 bioengineering-11-00255-f006:**
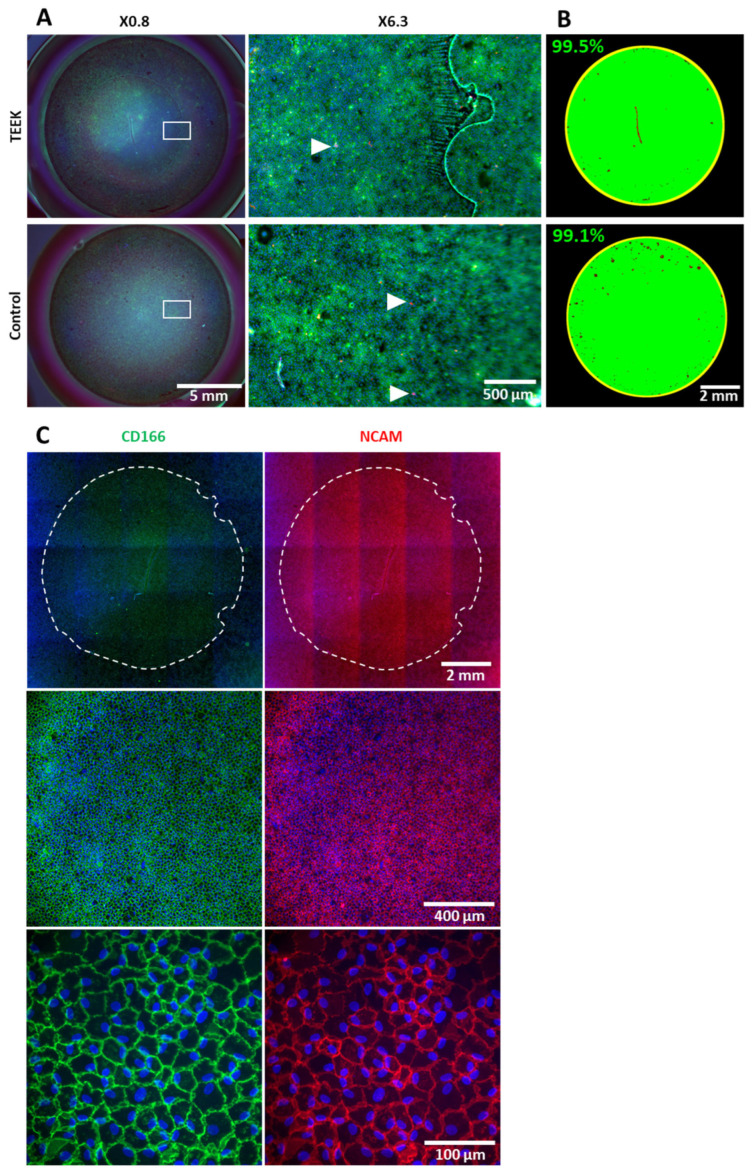
Viability and phenotype of corneal endothelial cells of TEEKs. (**A**) Viability/mortality assay with Hoechst 33,342 (blue), Ethidium homodimer (red), and Calcein-AM (green) of 4-week-seeded CECs on a lens capsule disc or directly on the plastic (control). Images are shown at low (×0.8) and intermediate (×6.3) magnification. White arrow shows dead cells (Ethidium+, Calcein-AM-). (**B**) Image analysis by ImageJ shows in green the surface covered with viable cells and in red the area of dead cells, on the whole TEEK and control surface. (**C**) Immunolabeling of CD166 and NCAM (CD56) counterstained with DAPI, at low, intermediate, and high magnification. The dotted line delimited the TEEK.

## Data Availability

The data that support the findings of this study are available from the corresponding author upon reasonable request.

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
