# Peer review of "Femtosecond Laser Cutting of Human Crystalline Lens Capsule and Decellularization for Corneal Endothelial Bioengineering"

_bioengineering, 2024, doi:10.3390/bioengineering11030255_

Round 1

Reviewer 1 Report (Previous Reviewer 2)

Comments and Suggestions for Authors

Femtosecond laser cutting of human crystalline lens capsule and decellularization for corneal endothelial bioengineering

Manuscript ID: bioengineering-2463008

This study reported the use of a femtosecond laser cutting process to obtain an 8 mm diameter lens capsule disc from an intact whole human intraocular lens. Additionally, a chemical-free decellularization method was employed in the study. The study is important in its field and well-constructed; however, there are several corrections that need to be addressed. Please refer to the following points:

  • Thank you for your comprehensive analysis of the manuscript. The responses provided have addressed most of the preliminary concerns succinctly. However, to advance the manuscript to the next stage, I must point out a significant shortfall in the current discussion. A more exhaustive and elaborate discussion is imperative, one that fully explores the implications and direct clinical applications in light of the study's results. Such a discussion is essential to fully understand the relevance and potential impact of the research. The authors are urged to enrich the discussion section with a thorough analysis of the study's impact on existing practices, its practical applications in clinical environments, and implications for future research. This expansion is crucial for deepening the manuscript's substance and offering a definitive guide on incorporating the research findings into current knowledge and clinical practice. To conclude, the allotted 800 words for the discussion section in the current journal format are inadequate to comprehensively address the study's implications, findings, and relevance to the field.
  • The authors have not indicated the corrections within the body of the manuscript in the file of the reply to the reviewer's text.
  • Observing the corrections in the article's latest draft, however, details for points 2 to 4 and point 6 are missing from the manuscript.
  • Furthermore, the new image referenced in point 3 is absent from the updated article.

Comments on the Quality of English Language

Minor to Moderate editing of English language required

Author Response

Reviewer 2 Report (New Reviewer)

Comments and Suggestions for Authors

The manuscript presents an interesting technique and I congratulate the authors on their research to improve tissue availability.

From a clinical point of view, there are some issues that should be addressed.

1-     tissue availability (tendency is for earlier cataract surgery and therefore less donors with their crystalline lenses)

2-     costs – femtosecond lasers for cataract surgery are quite expensive and not readily available

3-     how would the crystalline capsule behave inside the eye? Would it adhere to the stroma? It´s possible that it behaves differently once floating in the anterior chamber and decellularized. It is also possible that the endothelial cells may detach from the lens capsule over time.

The authors should address these issues or acknowledge that these issues should be studied in the future.

Minor comments:

1-     the sentence “If the corneal edema persists for only a limited duration, typically a few months at most, the restoration of endothelial function through endothelial transplantation may enable the complete recovery of corneal transparency” should be “If the corneal edema persists, the restoration of endothelial function through endothelial transplantation may enable the complete recovery of corneal transparency”, as even if there is corneal edema for years it is possible to restore corneal transparency with endothelial transplantation.

2-     The sentence “Endothelial graft is a surgical technique which has revolutionized the prognosis of corneal pathologies for the past 15 years…” should be “Endothelial transplantation has revolutionized the prognosis of corneal pathologies for the past 15 years”

3-     Page 2: “…a primary endothelial disease affecting a various degrees 4 to 10 % of adults…” should be “…a primary endothelial disease affecting IN various degrees 4 to 10 % of adults…”

4-     Page 2: “Secondly, complications of crystalline lens surgery called bullous keratopathy (the rate of endothelial decompensation is low but cataract surgery is the most frequent…” should be “Secondly, complications of crystalline lens surgery (the rate of endothelial decompensation is low but cataract surgery is the most frequent…”. Bullous keratopathy is the end stage of corneal edema, but many complicated lens surgeries lead to corneal edema without bullous keratopathy.

Comments on the Quality of English Language

some minor corrections

Round 2

Reviewer 1 Report (Previous Reviewer 2)

Comments and Suggestions for Authors

The authors have submitted a revised version of the manuscript, addressing all comments made by the referees. The article is now in compliance with the required standards.

Comments on the Quality of English Language

The authors have submitted a revised version of the manuscript, addressing all comments made by the referees. The article is now in compliance with the required standards.

This manuscript is a resubmission of an earlier submission. The following is a list of the peer review reports and author responses from that submission.

Round 1

Reviewer 1 Report

Comments and Suggestions for Authors

The manuscript presents a relevant study focusing on the production of scaffolds generated from anterior lens capsule for the bioengineered corneal endothelial grafts. The authors investigate the generation of specific circular discs, measuring 8mm in diameter, by lasering the anterior capsule of the crystalline lens using a customized femtosecond laser. These discs are subsequently decellularized.

While the manuscript provides a straightforward account of the research, it relies rather heavily on references to the same group's previous work for information regarding the functionality of the scaffold.

I do, however, have several concerns and suggestions for improvement.

Firstly, there are several sentences within the manuscript that would benefit from rephrasing to enhance readability and clarity. Specifically, attention should be given to Line 37-39, Line 40-42, Line 46-53, Line 103-104, Line 105-108, and Line 169-171.

Secondly, it is crucial for the authors to discuss the external scalability of their approach (to other centres) and to provide insight into the difficulties that external parties may encounter when attempting to reproduce the production of circular lens discs using such an experimental femtosecond laser. This consideration is vital to ensure the practicality and applicability of the proposed method.

Furthermore, while the authors refer to their previous work involving the use of 6mm diameter TEEK composed of similar anterior lens capsule discs seeded with human corneal endothelial cells in a preclinical study on rabbits, the current manuscript feels incomplete without the inclusion of data characterizing primary cells seeded onto the 8mm diameter discs produced using the femtosecond laser and decellularized as described in the manuscript. It is essential to present a comprehensive evaluation of the entire process, including seeding the discs with primary cells at a predetermined density and maintaining them for at least 3 to 4 weeks to confirm the sustainability of their cell density. 

Comments on the Quality of English Language

I believe it would be highly advantageous for the authors to have the final version of the revised preparation proofread by an individual with a strong command of the English language to ensure the flow and clarity of the manuscript.

Reviewer 2 Report

Comments and Suggestions for Authors

Manuscript ID: bioengineering-2463008 

This study reported the use of a femtosecond laser cutting process to obtain an 8 mm diameter lens capsule disc from an intact whole human intraocular lens. Additionally, a chemical-free decellularization method was employed in the study. The study is important in its field and well-constructed; however, the results section is lack in vital information and there are several corrections that need to be addressed.

Please refer to the following points:

  1. Although this is a short communication article, please add information in the results section about decellularization like histological analysis, biomechanical testing, cell quantification and DNA content.
  2. The authors should provide and addressed in detail about the acceptance normal size graft of a minimum diameter of 8.5 millimeters (and not 8 mm).
  3. Figure 1 (Lens holder and femtosecond laser pattern): It would be preferable to include a real photo of the device instead of an illustration.
  4. In the methods and materials section, please indicate the city and country of the study in parentheses.
  5. The chapter on anterior lens capsule dissection should be expanded starting from line 147.
  6. Survival of these grafts- need to be addressed.

Spelling/grammar:

Please review grammar/sentence structure of the entire manuscript to improve clarity of intent and flow of the manuscript. Some examples:

*Line 39

*Line 42

*Line 52-3

*Line 54 change "offer" to "supply"

*Line 56 change "from one country to another" to "between countries"

Comments on the Quality of English Language

Minor to Moderate editing of English language required